

# Rain gauges and X-band radar hourly comparison under complex orographic conditions in Reunion Island during the passage of the cyclone Batsirai

Ambinintsoa Volatiana RAMANAMAHEFA[1], Thiruvengadam PADMANABHAN[1,2], Guillaume
LESAGE[1] and Joël VAN BAELEN[1]

[1]Laboratoire de l'atmosphère et des Cyclones LACY, UMR 8105 CNRS, Faculty of Sciences and Technologies, Universié de
La Réunion, 97400 Saint-Denis, France
[2]School of Meteorology, University of Oklahoma, Norman, Oklahoma, United State

*Correspondence to*: Ambinintsoa V. RAMANAMAHEFA (ambinintsoa.ramanamahefa@univ-reunion.fr) and Joel VAN
BAELEN (joel.van-baelen@univ-reunion.fr)

**Abstract.** Weather radar observations and quantitative precipitation estimation (QPE) are in the early stages of development in the South-West Indian Ocean (SWIO) region, which is prone to heavy rainfall, particularly during the passage of tropical cyclones. Given the topography of SWIO islands, orography plays an important role in the spatial distribution of precipitation. The ESPOIRS project was designed to investigate such dynamics in Reunion Island, Seychelles, and Madagascar using a mobile X-band radar. Reunion Island served as a testbed to evaluate X-band radar retrieved QPE using specific comparisons between several radar approaches and available rain gauges. This is the first study to use an X-band dual-polarization radar in the SWIO region. Our research focuses on the intense tropical cyclone event Batsirai in Reunion Island and shows the effectiveness of dual-polarization radar when compared to single-polarization radar in mitigating reflectivity attenuation. Both the Hitschfeld and Bordan and the philinear algorithms were employed and evaluated for this purpose. As our study encountered challenges related to noisy and low-resolution differential phase ($\phi_{dp}$) data, we detailed the pre-processing steps used to extract reliable $\phi_{dp}$ data from the observed measurements. Furthermore, we tested two precipitation estimators, R(Z) and R(kdp). We observed that the accuracy of R(Z) depends on the attenuation correction method. Additionally, using the extracted $\phi_{dp}$, we calculated an empirical model for R(kdp) for Reunion Island. This model provided better results compared to the R(Z) estimates, which can be explained by the fact that kdp is directly linked to precipitation concentration and does not require attenuation correction. Our findings highlight that the accuracy of the radar QPE is strongly influenced by local topography, which in turn governs local rainfall patterns, while the accuracy of QPE also depends on the type of precipitation
.





## 1 Introduction

The South-West Indian Ocean (SWIO) islands have a tropical climate and often experience heavy rainfall, especially during the cyclonic season. For example, Reunion Island holds rainfall world records, including most accumulated precipitation in 12h, 24h and 72h during tropical cyclonic events due to the orographic effect on precipitation. For this reason, one focus of the ESPOIRS project, standing for "Study of Indian Ocean Precipitating Systems by Radar and Satellites" ("*Etudes des*

*Systèmes Précipitant* dans l'Océan Indien par Radar et Satellite" in French), is to study orographic precipitation with respect to the contrasting orography of Reunion Island, Seychelles, and Madagascar. Therefore, this study requires data with a high spatial and temporal resolution, as precipitation in mountainous regions varies greatly in space and time (Barros and Arulraj, 2020). To achieve this, a mobile doppler dual-polarization X-band radar with high spatial and temporal resolution was acquired and deployed successively in Reunion Island, Seychelles, and Madagascar.

40        Although X-band radars are affected by attenuation, they are still an excellent tool to obtain detailed observations (Yang et al. 2023; Antonini et al. 2017). However, radar-based quantitative precipitation estimation (QPE) is still in the early stages of development in the SWIO region. This is the first study on radar QPE using dual-polarization X-band radar in this region whose complex terrain makes the study challenging. Currently, in SWIO, only Reunion Island combines two S-band radars with a fairly dense network of rain gauges with 6 min and 1 h temporal resolutions. Therefore, for the ESPOIRS project,

Reunion Island served as an appropriate testbed to assess the quality of the QPE derived from the ESPOIRS X-band radar.

        For this paper, two estimators, R(Z) and R(kdp), were used for the radar QPE. However, reflectivity (Z) is affected by attenuation caused by liquid clouds and precipitation, especially in the X-band radar (Delrieu et al., 1999), which is more pronounced in tropical regions like SWIO. For example, at a 3-cm wavelength (X-band), echoes located behind a thunderstorm cell with a width of 5-10 km can be approximately 95% weaker than their strength if the cell had not been present (Fabry,

2017). The literature proposes various methods for correcting attenuation. Generally, correction methods for attenuation of radar reflectivity are based on two principles. The first set of methods is based on single polarization and requires only reflectivity as the input. It uses empirical relationships to calculate the specific attenuation A from reflectivity Z (Jacobi and Heistermann, 2016), such as the methods of Hitschfeld and Bordan (1954), Harrison et al. (2000), and Krämer and Verworn (2008). The advantage of these approaches lies in their ease of implementation, although they heavily depend on proper radar

calibration. The second set of methods is based on dual polarization, where attenuation is estimated from the differential phase shift ($\phi_{dp}$) (Bringi et al., 1990; Testud et al., 2000; Park et al., 2005). This approach provides more relevant rainfall data than the single-polarization method. However, since the radar does not directly measure the real differential phase shift $\phi_{dp}$, some data pre-processing is required to extract $\phi_{dp}$ from the observed $\Psi_{dp}^{obs}$, which can challenging (Fabry, 2017).

        The objective of this paper is twofold. First, it aims to evaluate the effectiveness of the single-polarization method

(Hitschfeld and Bordan, 1954) alongside dual-polarization correction methods, specifically the philinear method (Bringi et al., 1990), for correcting reflectivity Z attenuation. This evaluation seeks to highlight the strengths and limitations of these methods. Second, it aims to achieve accurate precipitation estimates using both R(Z) and R(kdp) estimators by considering the





impact of the terrain on each rain gauge location and the type of precipitation (stratiform or convective) on the radar quantitative precipitation estimation (QPE).

The paper is structured as follows. Section 2 provides a comprehensive description of the dataset, outlines data pre-processing, details the two methods used for reflectivity attenuation correction, namely the Hitschfeld and Bordan (1954) method and the $\phi$-linear method (Bringi et al., 1990), and details the estimation of R(kdp) specific to Reunion Island. Section 3 compares the radar-based quantitative precipitation estimates (QPE) with ground-based rain gauge measurements and discusses the findings. Finally, section 4 presents the conclusions and perspectives of this study.

## 70    2 Dataset description and meths

### 2.1 Study area and data

In the initial phase of the ESPOIRS project, a doppler dual-polarization X-band radar was deployed in Reunion Island ahead of the campaigns in Seychelles and Madagascar. Quality control of radar data was conducted in Reunion, where Météo France has set up an extensive network of rain gauges with a high temporal resolution of 6 min and 1 h. As the project focuses mainly

on heavy precipitation, this study was carried out with data collected during the intense cyclone Batsirai, which spanned from 1 to 4 February 2022, a period when continuous radar and rain gauges data were available.

The radar was located in Saint-Joseph in the south of Reunion Island (Figure 1b and 1c), at 20 m above sea level, with a maximum range set at 75 km. The radar performed 360-degree azimuthal scans along 12 constant elevation angles at 1.0°, 2.2°, 3.3°, 4.4°, 7°, 9°, 11°, 15°, 19°, 21°, 25°, and 29°, with a range gate resolution of 125 m. The complete set of scans was

repeated at 10 min intervals. Table 1 lists the main characteristics of the radar.

Six rain gauges were chosen to validate the radar data (Figure 1b and 1c), because they were located within the radar's field of view and provided continuous data recordings. The closest unobstructed radar beam to each rain gauge was selected to compare the rain gauge measurements with the corresponding radar estimate (Figure 2).






**(a)**

**(b)**

**(c)**

**Figure 1: a) Location of the South-West Indian Ocean; b) digital elevation model of Reunion Island showing the radar localization and nearby rain gauges; c) zoom on the area of interest**






**Table 1: Characteristics of the ESPOIRS radar**

| | |
|---|---|
| **Operating frequency** | 9410 MHz ± 30 MHz |
| **Peak power** | 25 kW (12.5 kW per channel) |
| **Transmitter** | Magnetron |
| **Pulse length** | 4 Pulse lengths within 0.25 and 1.1 μs |
| **Pulse repetition frequency** | 900 to 3000 Hz (within the duty cycle) |
| **Minimal range resolution** | 25 m |
| **Operational range** | Up to 150 km |
| **Polarization** | Dual polarization (H/V) |
| **Receiver** | Dual pol. (2 independent channels) / Doppler |
| **Antenna** | 1.3 m splash plate parabolic antenna |
| **Beam width** | < 2° |
| **Antenna motion** | Volume scan (multi-elevation PPI's and RHI capability) |
| **Maximum movement speed azimuth** | 36 °/s |
| **Maximum movement speed elevation** | 15 °/s |

Table 2 below presents the characteristics of the selected rain gauges based on their location with respect to the radar location and the selected radar beams, revealing two distinct groups of rain gauges:

- **Group 1:** rain gauges located within 15 km of the radar with a vertical distance between the radar beam and rain gauge of less than or equal to 600 m, including Crete, Grand-Coude, and Grand-Galet.


- **Group 2:** rain gauges located at distances greater than 15 km from the radar, with a vertical distance exceeding 800 m between the radar beam and rain gauge, including Tampon, Commerson, and Bellecombe.






**Table 2: Characteristics of the rain gauges and elevation angles of the considered radar beams**

| Stations | Group | Altitude above sea level (m) | Distance from radar (km) | Vertical distance between radar beam and rain gauge (m) | Elevation of radar beam (°) |
|---|---|---|---|---|---|
| Crête | 1 | 659 | 8 | ~350 | 7 |
| Grand-Coude | 1 | 1085 | 9.2 | ~400 | 9 |
| Grand-Galet | 1 | 505 | 8.6 | ~600 | 7 |
| Tampon | 2 | 860 | 16 | ~1 130 | 7 |
| Bellecombe | 2 | 2 245 | 20 | ~960 | 9 |
| Commerson | 2 | 2 310 | 19.6 | ~830 | 9 |
















**Figure 2: Location of radar and radar beam relative to the rain gauge locations**





## 2.2 Radar data processing

The processing applied to radar data is detailed in this section and summarized in Figure 3.

In the first processing step, all non-meteorological signals were removed using two filters. The first filter consists of
removing signals with copolar correlation coefficient $\rho_{HV}$ values less than 0.85. This simple but efficient tool for data quality control is an easy way to remove non-meteorological echoes, since in most precipitation regions, $\rho_{HV}$ values typically exceed 0.8 (Rauber and Nesbitt, 2018). The second filter involves using a horizontal signal-to-noise ratio (SNRH) greater than 3, a threshold value recommended by our radar manufacturer "Gamic" to suppress noisy data; higher values indicate better data quality.

In the second processing step, the attenuation of reflectivity Z was corrected by employing one of two algorithms: 1) an algorithm based on single polarization proposed by Hitschfeld and Bordan (1954), with the advantage of its ease of implementation as detailed in Section 2.2.1; and 2) an algorithm based on dual polarization, known as the philinear method, which uses the differential phase $\phi_{dp}$ and requires pre-processing, as detailed in Section 2.2.2 below. The respective performances of these two algorithms were compared with rain gauge measurements using the Z-R relationship defined by
Météo France as $Z = 300R^{1.35}$ for Réunion Island, with Z corrected for attenuation. An additional estimator was used to derive the radar QPE: $R = 8.06Kdp^{0.49}$, calculated from our radar observations and detailed in Section 2.2.3 below.

In this study, we also explored the influence of precipitation type on the radar QPE. Two classes of precipitation were therefore defined: the first relates specifically to stratiform precipitation, while the second includes all precipitation occurring during the study period by combining convective and stratiform precipitation.

According to the literature (Fabry and Zawadzki, 1995; Matrosov, 2021), stratiform rain is associated with the bright band. In tropical regions, stratiform rain often originates from old convective cells (Houze, 1997). The turbulence and vertical motion of convective rain inhibit the formation of a bright band (Ghada et al., 2022), which thus becomes a marker of stratiform rain. In our precipitation classification methodology, the identification of stratiform rain is based on the detection of the bright band in the maxdisplay plots. The bright band can be recognized by a circular region with elevated reflectivity values, observed
prominently in both the maximum reflectivity values around the radar and the maximum vertical cross-sections of the maxdisplay plots. Its presence signifies the occurrence of stratiform precipitation during the corresponding time step.

The agreement between radar-based QPE and rain gauge measurements was evaluated using the normalized bias (NB) and correlation coefficient (corr).


$$NB = \frac{\bar{R}}{\bar{G}} - 1 \tag{1}$$

$$corr = \frac{\sum_{\forall i}(G_i - \bar{G})(R_i - \bar{R})}{\sqrt{\sum_{\forall i}(G_i - \bar{G})^2}\sqrt{\sum_{\forall i}(R_i - \bar{R})^2}} \tag{2}$$





where R and G represent hourly rainfall observed by the radar and rain gauge, respectively, while $\bar{R}$ and $\bar{G}$ denote the average.




**Figure 3: Pre-processing chain for the radar quantitative precipitation estimation (QPE)**





### 2.2.1 Attenuation correction based on the single-polarization algorithm: Hitschfeld and Bordan (1954) method

To correct the attenuation of reflectivity (Z) due to precipitation, the Hitschfeld and Bordan (1954) method (HB method) was
one of the first to pioneer forward gate-by-gate procedures for attenuation correction. They used an empirical relationship
between specific attenuation A and reflectivity Z, represented as: $A = cZ^d$ (db/km) where c = 0.000149 and d = 0.757 at X-
band radar (Berne and Uijlenhoet, 2006). To correct the power loss of signals received at the $i$th range location, they used
reflectivity measurements at the previous ($i$ - 1)th range locations by accumulating the specific attenuation A of each gate. This
accumulation is referred to as path-integrated attenuation (PIA). $\text{PIA}_i$ in gate $i$ is expressed as follows:

$$\text{PIA}_i = c\left(Z_i + \sum_{j=0}^{i-1} cZ_j^d\right)^d \cdot 2\Delta r \quad (dB) \tag{3}$$

where $\Delta r$ represents the gate length.

Corrected reflectivity $Z_{\text{corr } i}$ (dB) in any range gate $i$ is obtained by adding the corresponding PIA to the uncorrected
reflectivity:

$$Z_{\text{corr },i} = Z_i + \text{PIA}_i \tag{4}$$

### 2.2.2 Attenuation correction based on polarimetric method: Philinear method

In this approach, PIA is estimated using the differential propagation phase shift $\phi_{\text{DP}}$ (Eq. 5). Bringi et al. (1990) demonstrated
that the attenuation experienced by propagating electromagnetic waves is directly proportional to the differential propagation
phase, serving as the fundamental physical principle behind attenuation correction in dual-polarization radar.

$$\text{PIA}(r) = \alpha \phi_{\text{DP}}(r) \tag{5}$$

$$Z_{\text{corr } i} = Z_i + \text{PIA}(r) \tag{6}$$

Eq. 6 is a simple method, commonly used in operational radars such as Météo-France (Figueras i Ventura et al., 2012).

$\phi_{\text{DP}}$ is immune to attenuation as long as the return power is above the noise power (Bringi and Chandrasekar, 2001)
and is unaffected by radar calibration (Bringi et al., 1990). According to Park et al. (2005), $\alpha$ in Eq. 5 depends on the air
temperature and drop shape. However, for radar frequencies above 9 GHz, temperature does not heavily affect attenuation, so
ignoring the temperature effects leads to only a minor increase in the error (Jameson, 1992). In the tropical region of West
Africa, $\alpha =$ 0.285 dB/°, a value calculated from observed radar data (Koffi et al., 2014). Météo France's operational setup
uses $\alpha =$ 0.28 dB/°, while our radar manufacturer Gamic applies the same coefficient to the X-band radar. Additionally, Yu,



Gaussiat, and Tabary (2018) found $\alpha = 0.276$ dB/°, a value that provides the best fit to minimize the bias in Z as a function of $\phi_{\mathrm{DP}}$. Therefore, in this study, we set $\alpha = 0.28$ dB/°.

### 2.2.2.1 Differential phase shift $\phi_{dp}$ pre-processing

The radar does not directly measure $\phi_{\mathrm{DP}}$; instead, it measures the total differential phase shift $\Psi_{dp}^{obs}$, expressed as:

$$\Psi_{dp}^{obs}(r) = \underbrace{\phi_{dp}(r)}_{1} + \underbrace{\delta_{co}(r)}_{2} + \underbrace{\phi_{dp_0}}_{3} + \underbrace{\sigma_{\phi dp}}_{4} \tag{7}$$

Where term 1 corresponds to the differential phase shift, term 2 to the backscattering copolar differential phase due to mie scattering, term 3 to the system differential phase offset, and term 4 to the standard deviation of the observation caused by the system fluctuation errors. In addition, factors such as partial beam blockage due to orographic effects can contribute to the existing uncertainty of the differential phase shift (Figueras i Ventura et al., 2012).

Eq. 7 reveals that the observed $\Psi_{dp}^{obs}$ exhibits several perturbation factors that necessitate pre-processing to estimate $\phi_{\mathrm{dp}}$ according to the five steps below:

1. Noise in $\Psi_{dp}^{obs}$ impacts the accuracy of the estimated $\phi_{dp}$. To address this issue, an additional filter was applied as follows. Initially, data are centred around 0 by removing the median of the data for the scan. Then, gradients are computed in both directions (azimuth and radius) to identify and eliminate data with high gradient values in both directions, as a high gradient value around a data point indicates noise. Finally, isolated pixels are removed from the dataset.

2. $\Psi_{dp}^{obs}$ can also experience folding in the event of heavy precipitation and appears when the phase shift exceeds 360° between the two measured polarizations (Rauber and Nesbitt, 2018). Moreover, the system differential phase offset $\phi_{dp_0}$ contributes to this folding, leading to an increase in $\Psi_{dp}^{obs}$. In our dataset, this folding mostly occurred as the cyclone approached Reunion Island.

3. The differential phase system offset $\phi_{dp_0}$ is calculated and removed. In theory, $\phi_{\mathrm{dp}}$ should start at 0° and increase with precipitation. To estimate $\phi_{dp_0}$, the first precipitation from the radar is determined. Our radar manufacturer Gamic proposed an algorithm to determine $\phi_{dp_0}$, which requires data with rain in the vicinity of the radar. This involves finding segments of precipitation close to the radar along each radius and calculating the median of $\phi_{dp_0}$ for each segment, corresponding to the first non-noisy values of $\phi_{\mathrm{dp}}$ in a ray. This process is iterated for each file to





ensure a consistent offset value for all the data. In other words, the final value of $\phi_{dp_0}$ is a single median value of $\phi_{dp_0}$ from several files. It is supposed to be constant regardless of the precipitation type, although it may vary between different sites (Frech, 2013). Figure 4 shows the offset values for each dataset and for each scan. The aim is to illustrate whether $\phi_{dp_0}$ changes depending on the situation.

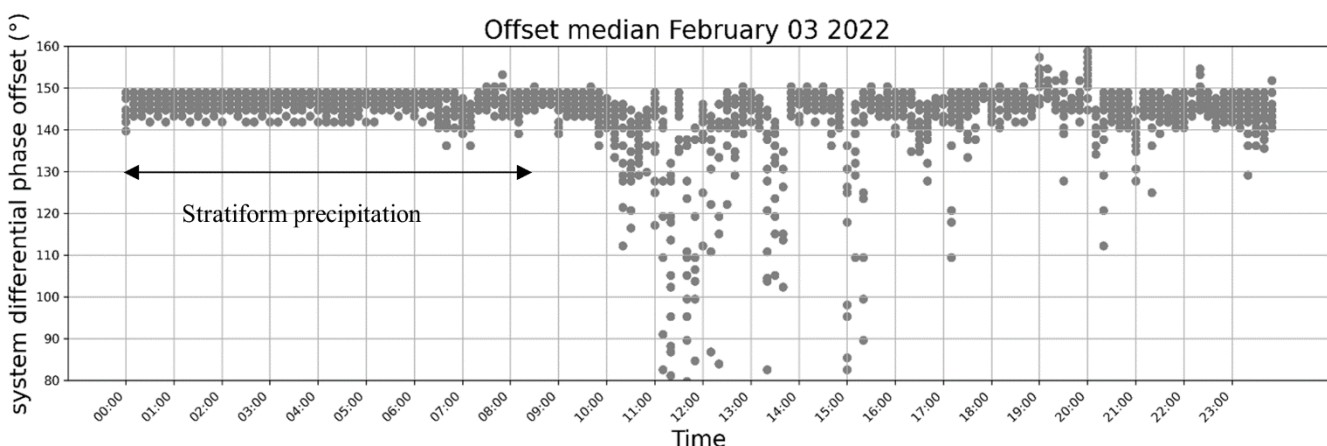

**Figure 4: Median of system differential phase offset for each scan and timestep on 3 February 2022 the period when the cyclone was close to Reunion Island (~190km from the north coast). During this period two types of precipitation were identified, stratiform precipitation from 00:00 to 8:40 UTC then the precipitation became heavy and characterized by convective rain.**

     Figure 4 illustrates that $\phi_{dp_0}$ can vary with time. This variation related to the type of precipitation. For instance, in the early morning until 8:40 UTC, the bright band was visible, indicating the presence of stratiform precipitation.

During this period, the offset value remained relatively stable. However, after 8:40 UTC, the bright band disappeared, with the precipitation becoming heavier with strong convective cells, resulting in variations in the offset value. Therefore, taking a single value of $\phi_{dp_0}$ for all the different cases can lead to errors in the attenuation correction. Figueras i Ventura (2012) pointed out that errors in $\phi_{dp_0}$ can lead to under- or overestimations of PIA, thus requiring a correction of the system's differential phase on a ray-by-ray basis. We thus tested a ray-by-ray method to improve

the $\phi_{dp_0}$ estimation using Py-ART, an open-source library (Helmus and Collis, 2016). Figure 5 shows the corresponding ray-by-ray $\phi_{dp_0}$ dependency at each elevation.




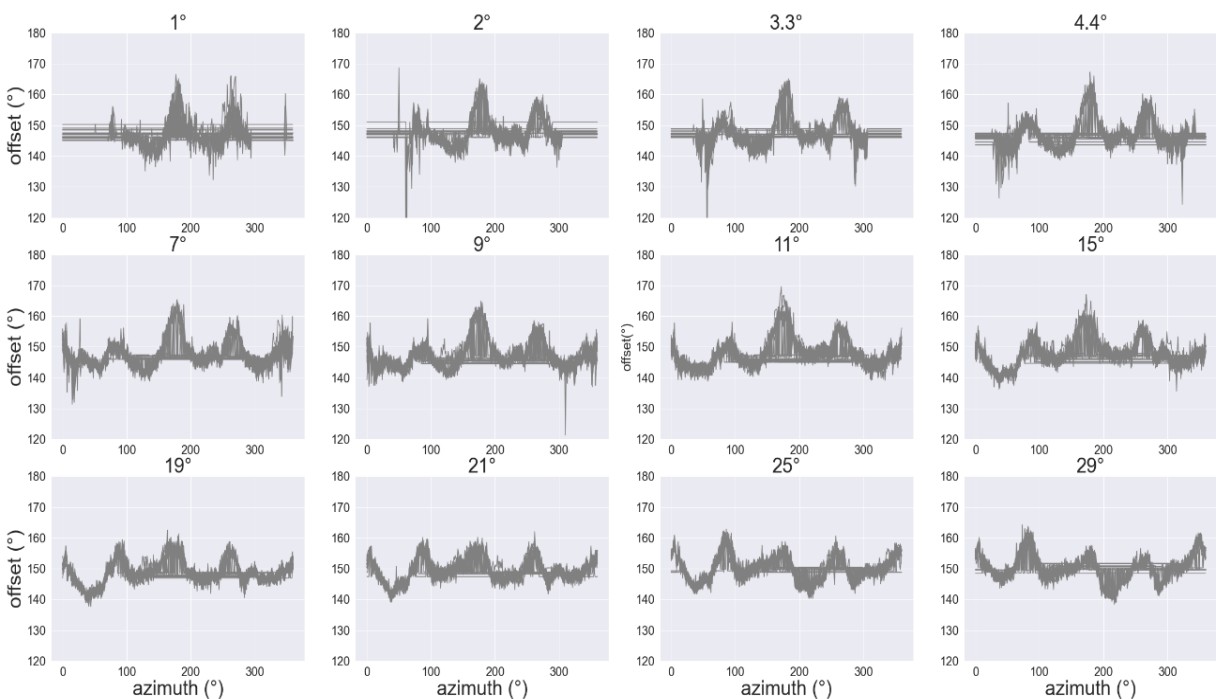

**Figure 5: Illustration of $\phi_{dp_0}$ on 3 February 2022 at 00:00 – 03:10 UTC ray-by-ray estimation during stratiform precipitation**

In all the panels of Figure 5, four peaks may be observed at the same azimuth location across the 360° azimuth scans except at lower elevations. This demonstrates the azimuthal dependence of the system differential phase offset, while at lower elevations, a significant portion of the radar beam is obstructed by the mountains over a large sector from approximately 300° to 90° azimuth, leading to disruptions in the observation of precipitation.

4. To attenuate the fluctuations of $\Psi_{dp}^{obs}$ caused by noises and a significant $\delta_{co}$ (Carey, 2000) associated with drop diameters above 2.5mm for the X-band radar (Trömel et al., 2013), the data were smoothed using a double-window smoothing technique available from the Py-ART library.

5. After these processes, an additional issue was identified: a persistent spurious signal along specific rays in $\phi_{dp}$. This anomaly is alleged to arise from the four radome joints (Figueras i Ventura, 2012). To mitigate this anomaly, we implemented the method detailed by Padmanabhan (2024).





Finally, Figure 6 illustrates an example of the estimated differential phase $\phi_{\mathrm{dp}}$ (black) from the total differential phase

shift $\Psi_{dp}^{obs}$ (blue).

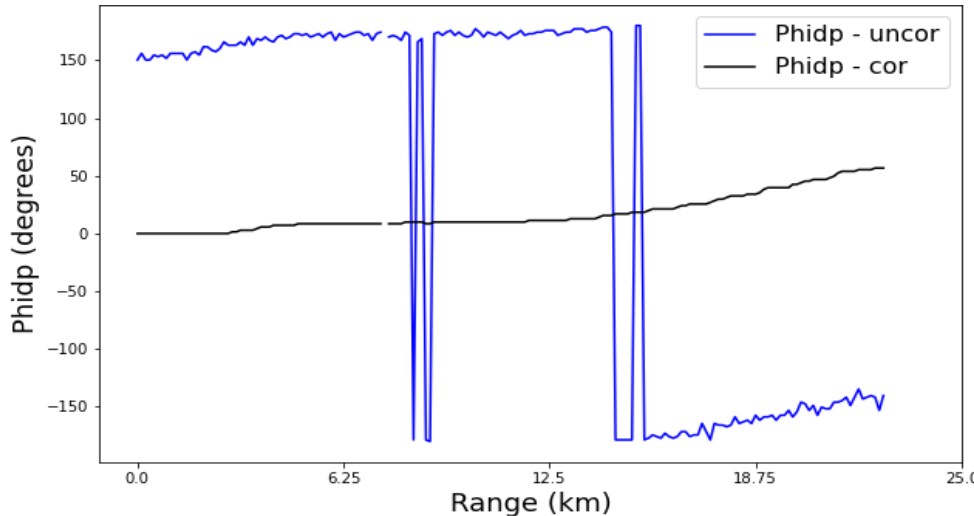

**Figure 6: Raw $\boldsymbol{\phi_{dp}}$ (blue) and corrected $\boldsymbol{\phi_{dp}}$ (black) for the scan at 9°, azimuth angle 337 on 3 February 2022 at 19:00 UTC**

**2.2.3 kdp and R(kdp) estimation**

The specific differential phase kdp is not directly measured by radar; instead, it is derived from $\phi_{\mathrm{dp}}$. The literature details

numerous methods for calculating kdp such as Vulpiani (2012), Maesaka (2012), Giangrande (2013), and Schneebeli (2014)

methods, with each method having strengths and weaknesses. Users should therefore choose the method best suited to their

data (Reimel and Kumjian, 2021). In this study, we used the Maesaka (2012) method, which estimates non-negative kdp and

manages significant kdp fluctuations in the case of weak rainfall, especially during stratiform precipitation. It provides an

accurate QPE for all rain intensities, even in weak precipitations without the need to use the Z-R relationship.

     After calculating kdp, the aim is to use it to estimate precipitation based on the relationship $R = aKdp^{b}$, where both

a and b are specific and depend on radar wavelength. To date, these values have not been calculated in the SWIO region.

Although our sample is limited, we used the least square fitting approach and obtained the relationship $R = 8.062Kdp^{0.4939}$,

as shown in Figure 7.




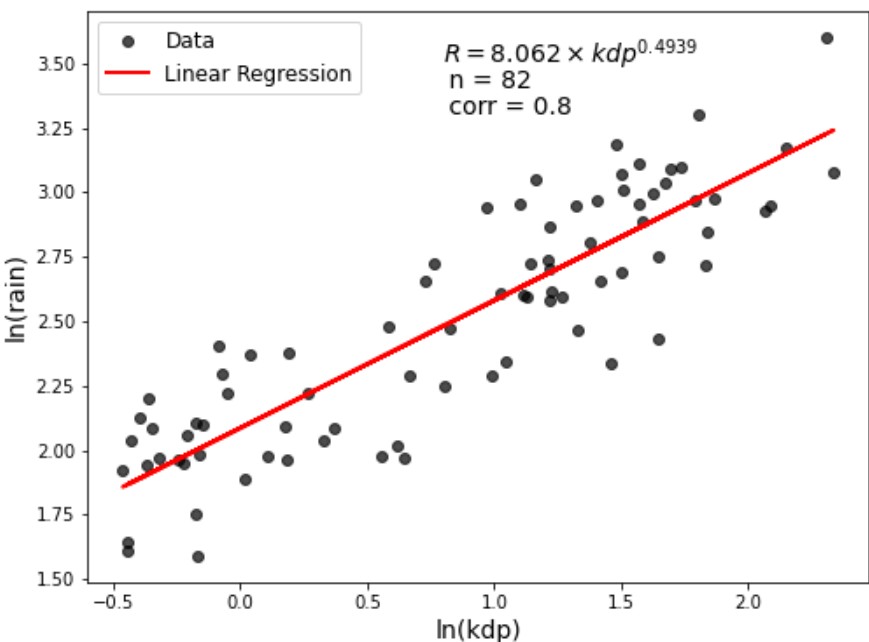

**Figure 7: R(kdp) estimated from radar observation and rain gauge measurements**


## 3 Results and discussion

Table 3 shows the correlation coefficient (corr) between the rain gauges and radar precipitation estimates, as well as the NB for each rain gauge location, each precipitation type, and each precipitation estimator R(Z) and R(kdp); n denotes the number
of samples. Additionally, Figure 8 provides an overall visualization of the radar QPE results. Subsequently, sections 3.1 and 3.2 present the overall results for the R(Z) and R(kdp) estimators. Section 3.3 further discusses the influence of precipitation type and orography on the radar QPE.







**Table 3: Global scores for radar quantitative precipitation estimation**

| Stations | Group | Rain rate from Z-R relationship $Z = 300R^{1,35}$ | | | | | | | | | | | | | | | | | Rain rate from R(kdp) $R = 8.062KDP^{0,4939}$ | | | | | |
|---|---|---|---|---|---|---|---|---|---|---|---|---|---|---|---|---|---|---|---|---|---|---|---|---|
| | | Hitschfeld and Bordan | | | | | | Philinear | | | | | | Estimation based on Reunion data | | | | | |
| | | All types of precipitation | | | Stratiform | | | All types of precipitation | | | Stratiform | | | All types of precipitation | | | Stratiform | | |
| | | corr | NB | n | corr | NB | n | corr | NB | N | corr | NB | n | corr | NB | n | corr | NB | n |
| Crête | 1 | 0.45 | 0.3 | 35 | 0.85 | 0.6 | 9 | 0.7 | 0.57 | 54 | 0.87 | 0.68 | 9 | 0.82 | 0.62 | 54 | 0.86 | 1.3 | 9 |
| Grand-Coude | 1 | 0.57 | 0.2 | 71 | 0.6 | 0.4 | 9 | 0.7 | 0.38 | 82 | 0.74 | 0.49 | 9 | 0.73 | 0.51 | 82 | 0.87 | 1 | 9 |
| Grand-Galet | 1 | 0.67 | 0.17 | 64 | 0.5 | 0.8 | 9 | 0.7 | 0.3 | 75 | 0.60 | 0.8 | 9 | 0.86 | 0.64 | 75 | 0.87 | 1.47 | 9 |
| Tampon | 2 | 0.57 | -0.5 | 16 | 0.8 | -0.15 | 6 | 0.8 | -0.4 | 31 | 0.82 | -0.2 | 6 | 0.78 | -0.1 | 31 | 0.87 | 0.3 | 6 |
| Bellecombe | 2 | -0.1 | -0.6 | 22 | 0.01 | -0.6 | 9 | 0.5 | -0.2 | 45 | 0.84 | 0.05 | 9 | 0.56 | -0.2 | 45 | 0.66 | 0.12 | 9 |
| Commerson | 2 | -0.1 | -0.6 | 39 | -0.2 | -0.67 | 9 | 0.4 | -0.2 | 57 | 0.2 | -0.2 | 9 | 0.65 | -0.2 | 57 | 0.29 | -0.13 | 9 |





**Figure 8: Radar quantitative precipitation estimation using a) R(Z) estimator and Hitschfeld and Bordan attenuation correction; b) R(Z) estimator and $\phi$-linear attenuation correction; c) R(kdp) estimator**





### 3.1 R(Z) estimator

In this section, the interpretation focuses solely on the 'all precipitation' types mentioned in Table 3, as the sample 'n' is sufficiently large.

395        According to Table 3, for all stations, the HB method showed a lower correlation coefficient than the philinear method, particularly for stations located more than 19 km from the radar, such as Bellecombe and Commerson (Group 2). At these distant stations, the HB method exhibited a negative correlation coefficient, suggesting that radar rainfall estimates and rain gauge measurements were not related.

        The NB reveals that for the rain gauges located less than 15 km from the radar (group 1) such as Crete, Grand-Coude

and Grand-Galet, both methods (HB and philinear) exhibit a positive NB, indicating a slight overestimation by the radar. Conversely, for stations located further from the radar (group 2) such as Tampon, Bellecombe, and Commerson, the NB is negative, indicating the radar's underestimation of precipitation. This underestimation is more pronounced for the HB method compared to the philinear method. For instance, for Commerson, the NB is -0.6 with the HB method and becomes -0.2 with the philinear method.

405        To summarize, the results show that the philinear method improves attenuation correction compared to the HB method. This improvement is due to the fact that the HB method relies solely on reflectivity as the algorithm input. Reflectivity is sensitive to attenuation, radar calibration errors, and the presence of non-rain scatter. Hitschfeld and Bordan (1954) emphasized that even a minor error in the radar calibration could result in significant inaccuracy in the rain rate measurements, particularly at 3 cm wavelength. These limitations negatively impact the rainfall estimation (Jacobi and Heistermann, 2016).

However, the use of the differential propagation phase shift directly related to path-integrated attenuation (PIA) as in the method helps to overcome these uncertainties (Zrnić and Ryzhkov, 1996), as $\phi_{dp}$ is immune to partial beam blockage, radar calibration, and ground clutter.

### 3.2 R(kdp) estimator

In this section, the interpretation focuses exclusively on the class "all types of precipitation" due to the sufficient number of

samples.

        For stations close to the radar (group 1), the correlation coefficient ranges from 0.73 to 0.86, indicating a relatively strong relationship between the radar QPE and the rain gauge measurements. However, the NB ranges from 0.51 to 0.64, showing the radar's slight tendency to overestimate precipitation.

        For stations located further from the radar (group 2), the correlation coefficient varies from 0.78 to 0.56, indicating a

weaker but still significant correlation. The NB ranges from -0.27 to -0.11, showing the radar's tendency to slightly underestimate precipitation in these cases.

        The results highlight the better radar rainfall estimation using R(kdp) compared with R(Z) for both groups of rain gauges. This is due to the fact that kdp is less sensitive to attenuation (Schneebeli 2014) and is closely linked to the rain





concentration within a radar volume. According to the literature, R(kdp) proves particularly effective at accurately estimating
heavy precipitation (Figueras i Ventura, 2012; Koffi, 2014; Maesaka, 2012). It is noteworthy that during this event, the median
rain rates exceed 10 mm/h, with a maximum of over 60 mm/h (Figure 9), thus highlighting the intense precipitation during
this cyclonic event.

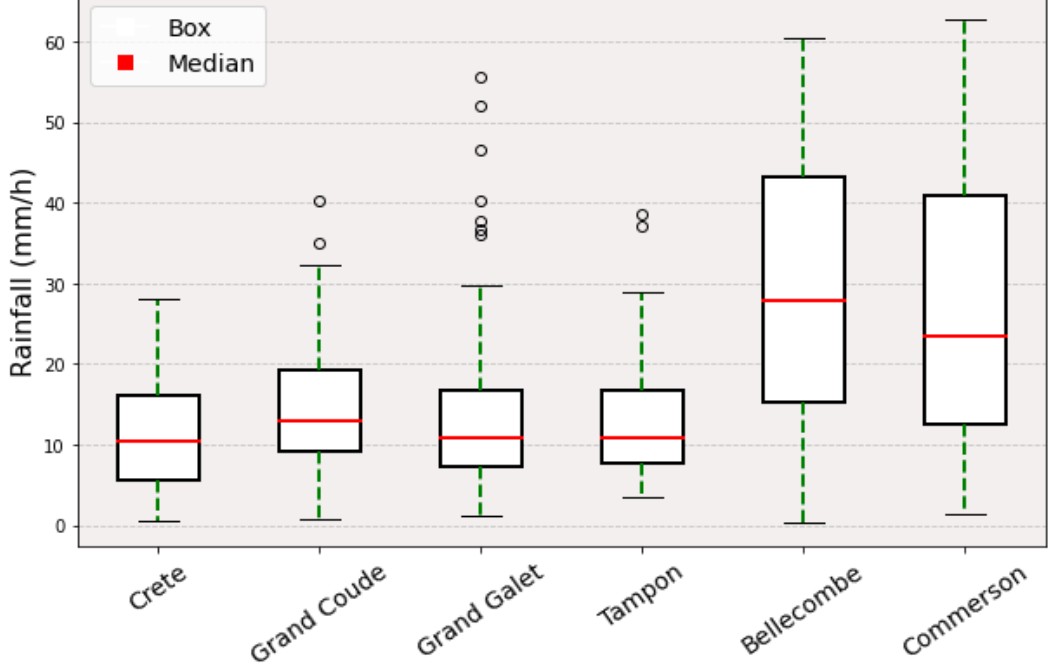

**Figure 9: Boxplot of hourly precipitation measured by rain gauges during the study period from 1 to 4 February 2022**

**3.3 Influence of precipitation types and orographic effects on the radar QPE**

According to Table 3, "stratiform precipitation" has a better score than "all types of precipitation" regardless of the distance
between the rain gauges and the radar or estimator used. One hypothesis is that stratiform precipitation is associated with
horizontally homogeneous rain (Sánchez-Diezma, 2000) and low turbulence, which explains the presence of the bright band.
Thus, despite the vertical separation between the radar beam and the rain gauges on the ground, and without considering
microphysical processes like evaporation, the precipitation measured within the radar beam at altitude has similar
characteristics as the rain drop reaching the rain gauges. The drop size distribution at this altitude will be similar as on the
ground.

As illustrated in Figure 10, a bright band may sometimes appear in certain regions (a) or be disrupted above others
(b), especially at higher altitudes. Ding (2014) revealed that precipitation types are significantly influenced by altitude. Terrain
and intense wind associated to the tropical cyclone cause air to flow over the mountains(Yu and Cheng, 2008), which leads to
produce more liquid water cloud (Lee et al., 2018) enhancing precipitation in higher elevation. It may explain why at





Commerson, located at 2,310 m above sea level, the correlation coefficient is lower, even during stratiform rain. Nevertheless, we should bear in mind that the limited number of stratiform precipitation samples limits the conclusions that can be drawn from the latter analysis.

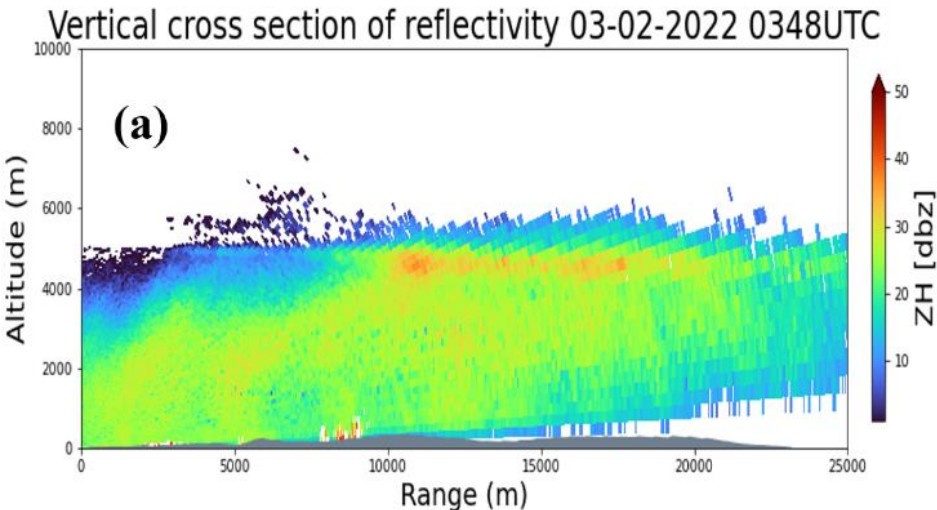

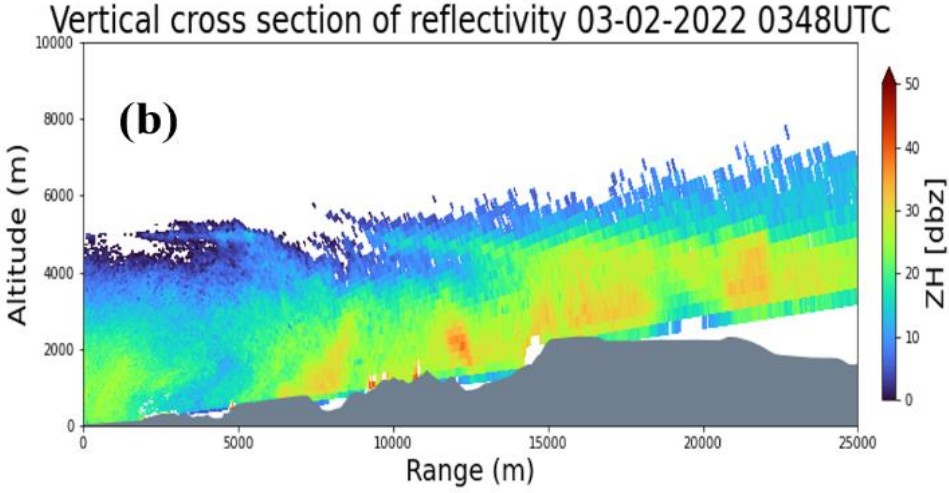

**Figure 10: Vertical cross section of reflectivity on 2 February 2022, at 3:48 UTC: a) towards the coastal region (RHI at 76° of azimuth), showing a bright band at about 5 km altitude; b) towards the mountains (RHI at 28° of azimuth), showing areas of increased reflectivity as the rain evolves with the altitude towards the ridges of salient orographic features**





To summarize, the comparison between the rain gauge measurements and the radar QPE can be classified as follows:

- Group 1 (including Crete, Grand-Coude, and Grand-Galet), vertically close to the radar beam (less than 600 m) and horizontally close to the radar (less than 10 km). As these stations are located close to the coast and at a middle altitude, rainfall intensity is relatively lower, with a median value of around 10 mm/h. At these stations, all estimators (i.e., R(Z) and R(kdp)) exhibit higher correlation coefficients and a positive NB. The impact of attenuation is less significant due
to their proximity to the radar.

- Group 2 (including Commerson, Bellecombe, and Tampon), located far from the radar beam at a distance exceeding 15 km, with the vertical distance separating the radar beam and the rain gauge being greater than 800 m. Although Tampon is in the same class as Bellecombe and Commerson, its results are different, as the radar QPE at Tampon is more closely correlated with the rain gauge observations. Tampon experiences a lower median precipitation rate that is
approximately half that of Commerson and Bellecombe, as depicted in Figure 9. As a result, because of the heavy rainfall, the scores for Bellecombe and Commerson are lower than the Tampon rain gauge scores. The radar signal is strongly attenuated, which explains why only the R(kdp) estimator works satisfactorily for these two stations.

Overall, various processes contribute to the disparity between the radar QPE and the rain gauge measurements, mainly due to the different sampling approaches of the two observations. The radar QPE provides real coverage over large volumes, while
rain gauges measure precipitation at specific points on the ground. However, this difference in sampling can introduce inconsistencies. Radar, for instance, samples a broader area but is affected by various factors like beam height and attenuation, while gauges give precise readings at ground level.

Furthermore, the drop size distribution may exhibit significant differences on the ground and at higher altitudes due to microphysical processes linked to the precipitation system dynamics and orography. Two factors may contribute to this
disparity. First, under the influence of horizontal wind, the drops detected by the radar just above the rain gauges may not reach the ground where the rain gauges are situated and may instead fall far away. Second, several microphysical phenomena can alter the drop size distribution, including coalescence, evaporation, and bursting of drops before they hit the ground. This discrepancy between the drop size distribution at altitudes and at ground level can impact the accuracy of precipitation estimates. Drop size distribution is one source of uncertainty in the radar QPE (Rosenfeld and Ulbrich, 2003). The values of
the a and b coefficients in the R(Z) relationship are based on different factors such as the type of precipitation (convective or stratiform) and the physical processes that influence the drop size distribution (Zeng et al., 2021).

Although rain gauge measurements are considered to be the "ground truth" for precipitation, they may also have associated errors (Dhiram and Wang, 2016), notably in high wind conditions.





## 4 Conclusion

The ESPOIRS project aims to investigate the interaction of intense precipitating systems with orography across different SWIO islands. As part of this project, an X-band radar was initially set up for just under a year in Reunion Island. Hence, quality control of the radar data was conducted using the available network of rain gauges from Météo France, which have high temporal resolution that allows for the comparison of quantitative precipitation estimates from radar with the rain gauge measurements. This comparison was carried out during the passage of cyclone Batsirai close to Reunion Island, which lasted

for 4 days and provided valuable insights into the use of X-band radar for the study of precipitation systems. The radar QPE was determined using two estimators: R(Z) and R(kdp). It was observed that the accuracy of the R(Z) relationship depends on attenuation correction method. For this study, two methods based on single and dual polarization were used: the Hitschfeld and Bordan (1954) method and the philinear method, respectively. It was observed that the HB method has difficulty in correcting attenuation in the case of heavy precipitation (median value exceeding 10 mm/h) and at a distance greater than 15

km from the radar. In this case, the radar tends to underestimate precipitation. The radar QPE is thus weakly correlated with rain gauge measurements. Using the philinear method, the results are better, regardless of whether the stations are located near or far from the radar. As a result, the philinear method was chosen to correct the attenuation of reflectivity Z for our entire dataset. However, under intense precipitation conditions, exceeding 20 mm/h of the median value, the accuracy of R(Z) estimates becomes constrained. Hence, kdp was directly used to estimate precipitation, thus significantly improving the results

when compared to R(Z), even for rain gauges located far from the radar at a distance exceeding 19 km, with heavy precipitation reaching up to 60 mm/h and high wind gusts of up to 159 km/h. This improvement is due to the independence of kdp regarding attenuation. In the process of our analysis, a specific R(kdp) relationship for Reunion Island was thus determined.

The discrepancies between radar-estimated rainfall rates and rain gauge measurements can be explained by the following factors.

1. The different altitudes of the rain gauges and radar beams can have a considerable effect. The precipitation measured by the radar situated above the rain gauges may not reach the rain gauges. This means that the drop size distribution of rainfall can not be the same at ground level as it is at the altitude where the radar sampled it. Note that R(Z) and R(kdp) relationships are influenced by the drop size distribution.

2. The relationships used to correct attenuation and estimate rain rates depend on several parameters, including rain

drop size distribution, radar frequency, and even ambient temperature. In our case, we only have information about radar frequency, which forces us to rely on empirical values from the literature, thereby limiting the precision of our radar QPE.

The complex terrain of Reunion Island adds significant interest to this study. The unique characteristics of each station, whether situated in a valley or on top of a mountain, influence the precipitation dynamics and thus play a significant role in the



differences observed with the radar QPE. Furthermore, the type of precipitation, whether stratiform or not, influences the radar QPE. However, the limited sample size of the rain event prevented us from further investigating this aspect.

This study highlighted the importance of X-band polarimetric radars for estimating precipitation during cyclonic events in Reunion Island, thus showing X-band polarized weather radars to be a useful tool to investigate and monitor precipitation in the SWIO island territories. The data processing techniques developed in this research will be extended to the

Seychelles and Madagascar datasets.

**Data availability:** All radar data are available at https://geosur.osureunion.fr/thredds/catalog/researchprogram/espoirs/1-Saint_Joseph/RADAR/Data/az-vol-75-0125-1to25deg-2022-01/2022/02/catalog.html. For information on the rain gauge data, please send requests to ambinintsoa.ramanamahefa@univ-reunion.fr or visit the Météo France website https://portail-api.meteofrance.fr/web/fr/api/DonneesPubliquesObservation


**Code availability:** The code is available upon request from the authors.

**Author contributions:** Ambinintsoa Ramanamahefa wrote the paper and implemented the data analysis. Thiruvengadam Padmanabhan, Guillaume Lesage, and Joël Van Baelen contributed to the implementation of the methodology and revised the

paper. Joël Van Baelen led the ESPOIRS project and radar field deployments.

**Competing interests**: The authors declare that they have no conflicts of interest.

**Acknowledgments:** This work is part of the INTERREG V ESPOIRS project (Study of Precipitating Systems in the Indian Ocean by Radar and Satellites). The ESPOIRS scientific program is led by LACy (University of La Réunion / CNRS / Météo

France) and funded by the European Union (FEDER program - GURDTI/20201589-0021087), Réunion Region, SGAR-Réunion, French State (CPER), and University of La Réunion.

We express our gratitude to Dr Nan Yu for sharing valuable insights during the discussion on kdp processing and attenuation correction. Additionally, we would like to thank Gamic for providing us with the algorithm to determine the system differential phase shift.



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
