# Peer review of "Rain gauges and X-band radar hourly comparison under complex orographic conditions in Reunion Island during the passage of the cyclone Batsirai"

_Atmospheric Measurement Techniques, 2024_

## Author Comment (AC1)

**RC2:**

This manuscript deals with the quantitative estimation of rainfall in the south-west Indian Ocean, more particularly on Réunion Island where the orographic enhancement of rainfall can be very significant. A polarisation diversity X-band research radar was installed and tested. The authors state that this is the first search radar of this type to be deployed in this region of the world. The authors chose as a case study the passage of cyclone Batsirai in February 2022. They propose to compare the rainfall estimates obtained by radar using data from 6 rain gauges available on the island as a reference. Two radar rain intensity estimators were tested: the first based on the radar reflectivity factor Z, the second on the specific differential phase Kdp. For the first estimator, which is very sensitive to attenuation by precipitation, two attenuation correction methods were applied and evaluated: the method of Hitschfeld and Bordan (1954) and the so-called phi-linear method of Bringi et al. (1990). The authors confirmed the results of the literature on the subject: the fact that the second correction method is more reliable than the first and that the estimate with Kdp is more efficient for intense precipitation than the Z estimator. A third method, directly deriving the rainfall intensity from Kdp is also used and compared.

This article does not provide any new knowledge per se, but reports on the first quantitative rainfall estimates using polarisation diversity search radar in the south-west Indian Ocean. Section 2 is interesting because it provides a detailed description of the differential phase shift pre-processing.

In my opinion, some points in the article need to be corrected or clarified, in particular certain equations. It is a pity that the operational radar data available from the National Meteorological Centre is not used (or that its use is not discussed) in this study. In addition, the study only covers one case study, which does not provide a robust assessment. The case chosen corresponds to a very specific cyclone situation. This is both a strength (the type of event is poorly documented, particularly the interaction with the terrain) and a weakness (there are doubts about the quality of the reference intensities provided by the rain gauges under these conditions, which makes it more difficult to compare radar and rain gauge estimates).

We thank the reviewer for the time devoted to evaluating our manuscript. The comments provided are highly valuable and have helped us to improve the clarity and accuracy of the work.

**Questions and suggestions:**

- **Page 2, Line 43: Are the S-band radar data usable? Could they have been used?**

S-band radars are operational in Reunion and the data are usable. However, the operational radar in southern Réunion (Piton Viler) is located at an altitude of 1,700 m, and the ESPOIRS radar at an altitude of 20 m. So, there is no measurement of operational radar bellow 1700m of altitude. In addition, the coverage of the operational radar is significantly affected by terrain blocking (see figure below). In our area of interest (southern Réunion), the radar beam is masked by the terrain. The mountain peaks, where precipitation has been most intense and where reflectivity attenuation is particularly significant, are mainly masked by more than 50%. These are the reasons why we rely exclusively on comparisons with rain gauges.

[Figure]

*Figure 1.2: Mask of reflectivity (%) from two operational S-band radars. The primary purpose of these radars is to monitor tropical cyclones approaching Réunion Island.*

- **Page 2, Line 53: The reference to Krämer and Verworn is only a conference paper, and therefore not peer-reviewed. Does it really provide information that would not be available in peer-reviewed articles published in peer-reviewed journals?**

Indeed, in this paper, we cited the first version from conference. the DOI for the published paper is https://doi.org/10.2166/wst.2009.282, we will change this reference on revised manuscript.

- **Page 2, Line 58: define the variable (total differential phase shift) and make a reference to section 2.2.2.1**

Well noted, thank you for the suggestion.

- **Page 2, Line 62: Isn't it too ambitious and sufficiently robust to reproduce this R-KDP law on a single case study?**

We acknowledge that the sample considered in this study is limited. When applying the $R(K_{dp})$ relationship from Beard and Chuang (1987), we observed an overestimation of rainfall. We hypothesize that this bias may be related to the specific characteristics of tropical cyclones, where the raindrop size distribution (DSD) differs from the climatological values used to derive the original $R(K_{dp})$ coefficients. This motivated us to reproduce $R(K_{dp})$ relationship based on our own observational data.

*(We have provided a detailed response in Reviewer 1's comment, as follow:*

*At the outset of our study, we tested the standard relationship, because the R(kdp) for X-band radar for South West Indian Ocean has not yet been calculated:*

$$R = c' \left( \frac{K_{dp}}{f} \right)^d \tag{1}$$

*where f is the frequency of the radar in GHz (f = 9.40Ghz for the ESPOIRS Radar). We applied the parameters c' = 129 and d = 0.85, which are derived from drop equilibrium shape distributions (Beard and Chuang, 1987)and have been validated for long-term French polarimetric radars (Figueras I Ventura et al., 2012).*

*Equation 1 becomes:*

$$R = 19 \left( K_{dp} \right)^{0.85} \tag{2}$$

*However, our results showed a clear overestimation with the radar-derived rainfall rates R(kdp), even for Group 2 rain gauges, where the highest rainfall intensities were recorded during the tropical cyclone.*

*Therefore, we calculated the coefficient of R(Kdp) from our radar observations. However, as noticed by the reviewer, the derived coefficients (a = 8.062, b = 0.49) fall outside the expected range for X-band radars (a = 14–20, b = 0.73–0.85) reported in Ryzhkov and Zrnić (2019). We acknowledge this discrepancy and provide the following explanations:*

*The coefficients of R(Kdp) relationships are highly sensitive to DSD characteristics, which vary significantly by precipitation regime (Unuma et al., 2025). Tropical cyclones exhibit distinct DSD properties compared to climatological rainfall: i) high concentrations of small and midsize raindrops; ii) relatively few large raindrops (exceed rarely 4mm); iii) elevated drop number concentrations at given reflectivity levels. These features have been documented in observational studies of Atlantic tropical cyclones (Tokay et al., 2008).*

$K_{dp}$ *is strongly influenced by both the size and number of drops (Timothy et al., 1999).. Higher drop concentrations lead to larger* $K_{dp}$ *values for a given rainfall rate.*

*Therefore, the interpretation of R(kdp) coefficients proposed by this paper is as follow:*

➢ *Lower coefficient a (8.062 vs. 14–20): high drop concentrations producing higher Kdp during tropical cyclone.*
➢ *Lower exponent b (0.49 vs. 0.73–0.85): a weaker power-law relationship, indicating less sensitivity of R to Kdp changes, which may reflect the more uniform DSD characteristics in tropical cyclone precipitation.*

*Unfortunately, no disdrometer measurements were available during this event, we could not investigate the DSD.*

*We fully acknowledge the limited scope of our empirical derivation. Ideally, a comprehensive investigation across multiple tropical cyclone events would strengthen the robustness of TC-specific R(Kdp) relationship. However, the logistical constraints of the mobile radar deployment during this project precluded such extensive sampling (detailed bellow, P.8).*

*This finding opens promising research avenues for investigating tropical-cyclone-specific* $R(K_{dp})$ *relationships for X-band radar, using disdrometer data and T-matrix scattering simulations. These theoretical calculations, based on observed TC-specific DSDs, provide an independent verification of* $K_{dp}$ *behavior.)*

- **Page 5, line 100-105, table 2: It is preferable to specify how beamwidth is defined: 3 dB? 6 dB?**

  The beamwidth is defined with 3dB. This will specified in the revised version.

- **Page 7, figure 2: first sub-figure "Crête", the value of the vertical distance between radar beam and rain gauge should be 350 m and not 659 m. 659 m" should be replaced by "350 m" to be consistent with Table 2.**

  Well noted, we will correct it in revised version

- **Page 8, Lines 170, 172 and figure 3: make the information consistent. The text indicates phi_dp<0.85 whereas the figure indicates phi_dp >0.85. However, it is consistent for and SNRH >3 in both the text and the figure.**

  Thank you for pointing that out. We will correct it.

- **Page 8, Line 180: separate "to" and "derive"; Page 10, Ligne 231: « db » to be replaced by « dB »**

  Well noted and thank you, it will be corrected for the revised version.

- **Page 10, Line 231: give the unit of Z here (is it dBZ or mm-/m3 ?)**

  The unit of $Z$ is $(mm^6 m^{-3})$.

- **Page 10, line 230-232: The equation and the values of c and d are not mentioned in the proposed reference of Berne and Uijlenjoet (2006). Please cite a reference that use this expression and these values. Moreover, usually, these coefficients c and d correspond to a relationship between specific attenuation (dB/km) and reflectivity factor expressed in mm6/m3 (and not in dBZ). Please mention clearly the units of each variable in all the equations.**

  This value was taken from Tridon(2011), page 125, first paragraph), where he reports alpha = 112000 and beta = 1.32, originally from (Berne and Uijlenhoet, 2006).

  The relationship between reflectivity factor $Z[mm^6\ m^{-3}]$ and specific attenuation $A[dB\ km^{-1}]$ is written as

$$Z = \alpha A^\beta \tag{1}$$

  From equation 1:

$$A = \left(\frac{Z}{\alpha}\right)^{\frac{1}{\beta}} = \alpha^{-\frac{1}{\beta}} Z^{\frac{1}{\beta}} \tag{2}$$

  Hitschfeld and Bordan (1954) express the specific attenuation as a power law of the form:

$$A = aZ^b \tag{3}$$

By comparing equations 2 and 3, the coefficients a and b are expressed as:

$$b = \frac{1}{\beta} \quad ; \quad a = \alpha^{-\frac{1}{\beta}}$$ (4)

Since $\beta = 1.32$ and $\alpha = 112000$ as mentioned above:

$$b = 0.757$$
$$a = 1.49 \times 10^{-4}$$

- **Page 10, Line 235: I think relationship (3) is wrong. The prefactor c and the exponent d would be applied twice, which seems wrong to me.**

  We identified this mistake after submitting the manuscript, and we thank the reviewer for pointing it out. The corrected equation is provided below. However, this correction does not affect our results because the processing was performed using the open-source implementation in wradlib, which already uses the correct formulation.

  https://docs.wradlib.org/en/2.2.0/generated/wradlib.atten.correct_attenuation_hb.html#wradlib.atten.correct_attenuation_hb

  $$PIA = \sum_{i=1}^{N} 2A_i \Delta r$$ (5)

  Where N is the number of the gate, $\Delta r$ is the radial gate length or range resolution in [m], for each gate $i$ :

  $$A_i[\,dB\ km^{-1}] = aZ_i^b$$ (6)

  and $Z_i[mm^6\ m^{-3}]$ is the reflectivity factor.

  Thus, the full expression becomes:

  $$PIA\,[dB] = \sum_{i=1}^{N} 2\big(aZ_i^b\big)\Delta r$$ (7)

- **Page 10, Line 245: Equation (6) is exactly the same as equation (4), so is it necessary to duplicate it?**

  Indeed, it is the same equation, and we have taken this suggestion into account for the revised version of the article.

- **Page 10, Line 258: "mie" -> "Mie".** Well noted, thank you

- **Page 14, Line 331-334: It is clear that only one event (cyclone Batsirai in early February 2022) is used to determine this R-Kdp relationship. This is very little to ensure a minimum of robustness. Moreover, it gives the impression that rain gauge data are used both to establish this relationship and to assess it. This raises the question of the independence between the rain gauge data used to establish the Kdp-R relationship and those used to evaluate the quantitative estimates made using this relationship: has the dataset been split in two to carry out these two operations?**

We thank the reviewer for this important question. To ensure the independence of the data, we split it into training and test datasets.

The calculation of $R(K_{dp})$ was performed as follows:

The rain-gauge data have a 6-minute temporal resolution, whereas the radar $K_{dp}$ estimates are available every 10 minutes. To ensure a consistent comparison, both datasets were aggregated to a common 30-minute temporal resolution. A 30-minute interval was retained only if valid measurements were available for both rainfall and $K_{dp}$.

Since $K_{dp}$ values close to zero may correspond either to light precipitation or to unreliable estimates, only strictly positive $K_{dp}$ values were considered. Any interval containing missing or non-positive $K_{dp}$ values was discarded. After applying these filters and separating the data into training and test sets, only 82 samples met the quality criteria and were included in the analysis.

- **Page 15, figure 7: Put the units of the variables on the axes.**

[Figure]

*Figure 2.2: R(kdp) estimated from radar observation and rain gauge measurements*

- **Page 16, table 3: What is the time step used? "Hourly" is in the title of the paper and mentioned for fig. 9 but should be also indicated in the text of section 3 and in the title of table 3 if it is this same timestep that is used.**

  The time step used is hourly, we will provide it in the text

- **Page 18, line 398: "the HB method exhibited a negative correlation coefficient, suggesting that radar rainfall estimates and rain gauge measurements were not related". It is rather the fact that the correlation coefficient is close to 0 that indicates an absence of link.**

  We will updated the manuscript accordingly, thank you for it.

- **Page 19, figure 9: What do you mean by " box " in the legend? Could you indicate which quantiles represent the limits of the boxes and whiskers in the boxplots?**

  The box spans the first (Q1) and third (Q3) quartiles, and whiskers extend from Q1 − 1.5×IQR to Q3 + 1.5×IQR, where IQR = Q3 − Q1.

- **Page 20, line 443: We can also imagine that there is a lot of wind, which must lead to a lot of errors in rain gauge measurement**

  We agree that the effect of wind in mountainous regions is not negligible and can contribute to measurement errors in rain gauges

**References**

Beard, K. V. and Chuang, C.: A New Model for the Equilibrium Shape of Raindrops, J. Atmospheric Sci., 44, 1509–1524, https://doi.org/10.1175/1520-0469(1987)044%253C1509:ANMFTE%253E2.0.CO;2, 1987.

Berne, A. and Uijlenhoet, R.: Quantitative analysis of X-band weather radar attenuation correction accuracy, Nat. Hazards Earth Syst. Sci., 6, 419–425, https://doi.org/10.5194/nhess-6-419-2006, 2006.

Figueras I Ventura, J., Boumahmoud, A., Fradon, B., Dupuy, P., and Tabary, P.: Long-term monitoring of French polarimetric radar data quality and evaluation of several polarimetric quantitative precipitation estimators in ideal conditions for operational implementation at C-band, Q. J. R. Meteorol. Soc., 138, 2212–2228, https://doi.org/10.1002/qj.1934, 2012.

Timothy, K. I., Iguchi, T., Ohsaki, Y., Horie, H., Hanado, H., and Kumagai, H.: Test of the Specific Differential Propagation Phase Shift (KDP) Technique for Rain-Rate Estimation with a Ku-Band Rain Radar, J. Atmospheric Ocean. Technol., 16, 1077–1091, https://doi.org/10.1175/1520-0426(1999)016%253C1077:TOTSDP%253E2.0.CO;2, 1999.

Tokay, A., Bashor, P. G., Habib, E., and Kasparis, T.: Raindrop Size Distribution Measurements in Tropical Cyclones, Mon. Weather Rev., 136, 1669–1685, https://doi.org/10.1175/2007MWR2122.1, 2008.

Tridon, F.: Mesure des précipitations à l'aide d'un radar en bande X non-cohérent à haute résolution et d'un radar en bande K à visée verticale. Application à l'étude de la variabilité des précipitations lors de la campagne COPS, 199, 2011.

Unuma, T., Yamauchi, H., Kato, T., Umehara, A., Hashimoto, A., Adachi, A., and Nagumo, N.: Characteristics of Raindrop Size Distribution Using 10-year Disdrometer Data in Eastern Japan, J. Meteorol. Soc. Jpn. Ser II, 103, 219–232, https://doi.org/10.2151/jmsj.2025-011, 2025.

---

## Author Comment (AC2)

**Review of "Rain gauges and X-band radar hourly comparison under complex orographic conditions in Reunion Island during the passage of the cyclone Batsirai**

*General thoughts and comments:*

The paper provides a description of a quantitative precipitation estimation data processing chain for an X-band radar deployed on Reunion Island. The most novel aspect of the paper is the location, with this being the first deployment of an X-band radar in the South-West Indian Ocean (SWIO) region. The paper then compares QPE from the radar to 6 rain gauges for a 4 day event, with dual-polarisation based techniques producing QPEs which compare better with rain gauges than the older, H-B technique of attenuation correction.

> **1- One of the key features missing from the paper, is the inclusion of an example PPI or two showing the radar coverage of the cyclone passage. A main advantage of weather radars is the wide area observations they provide, and it is difficult to understand how the X-band is benefitting Reunion without some representative figures.**

First, we would like to sincerely thank the reviewers for the time and effort dedicated to evaluating our paper. Their insightful comments and suggestions have been extremely helpful, and we believe they have significantly improved the clarity and overall quality of the paper. We also wish to thank the editor for handling our paper and for giving us the opportunity and time to revise.

Réunion Island operates two S-band operational radars, which are essential for monitoring tropical cyclones approaching the island. However, because these radars are installed on mountain summits, they cannot observe what happens at lower altitudes.

This limitation motivated the installation of an X-band radar, as part of ESPOIRS project, in the south of Réunion. The main objective is to study how precipitation interacts with the island's steep orography.

In Figure1, we have added the track of TC Batsirai together with an example of 11° PPI raw reflectivity scan. We selected the 11° elevation because it is the lowest elevation that remains unaffected by beam blockage from the surrounding mountainous terrain, thus providing the most reliable horizontal coverage. We provide also 2 examples of PPI scans.

[Figure]

Figure 1: (a) Location of the South-West Indian Ocean and track of the tropical cyclone Batsirai (TC); (b) Digital elevation model of Réunion Island showing the radar site, nearby rain gauges, the 75-km radar range, and the 20-km radius within which the radar beam is the closest (vertically) to the rain gauges. (c) Zoom over the study area. (d) PPI of raw reflectivity at 11° elevation, observed at 19 UTC on 3 February 2025.

[Figure]

*Figure 2: Appendix 1: (a) PPI reflectivity at 11° elevation when the TC center was located off the northeast coast of Réunion Island (00:00 UTC, 3 February 2022). The panel on the right shows a zoom over southwestern Réunion within a 20-km radius. (b) Same as (a), but when the TC canter was located off the northwest coast of Réunion Island (18:10 UTC, 3 February 2022).*

2- **Another limitation is the implementation of R(Kdp). The authors derive a relationship based on an empirical fit to the Sdata in the case study, which is very limited in scope. A quick comparison to the R(Kdp) relations detailed for X-band radars in Ryzhkov and Zrnic (2019) suggests the coefficients derived fall outside the expected range, with an a of 8.062 comparing to values between 14 and 20 and a b of 0.49 comparing to a range between 0.73 and 0.85. The authors do not comment on these differences and whether they are a result of the derivation methodology, which we have to assume uses aggregated radar data (6-min frequency) to compare with hourly gauge totals potentially for a single range gauge site (based on a n value of 82 in Figure 7, which matches the n value for Grande-Coude in table 3) or due to the unique drop-size distributions observed during the cyclone. At a minimum the authors should expand their description of the empirical derivation to explain how the temporal resolution issues are accounted for and exactly which gauge data is used. The authors should also consider how R(Kdp) performs when using a**

**representative, disdrometer derived, relation from the literature that is as close as possible to their case study (tropical cyclone) as that would be more applicable. If disdrometer observations are available from Reunion they could provide a third avenue to improving the use of Kdp in this study.**

We thank the reviewer for this insightful comment regarding the R(Kdp) relationship and the comparison with established coefficients in Ryzhkov and Zrnić (2019).

At the outset of our study, we tested the standard relationship, because the R(kdp) for X-band radar for South West Indian Ocean has not yet been calculated:

$$R = c' \left(\frac{K_{dp}}{f}\right)^d \qquad (1)$$

where f is the frequency of the radar in GHz (f = 9.40Ghz for the ESPOIRS Radar). We applied the parameters c′ = 129 and d = 0.85, which are derived from drop equilibrium shape distributions (Beard and Chuang, 1987) and have been validated for long-term French polarimetric radars (Figueras I Ventura et al., 2012).

Equation 1 becomes:

$$R = 19\left(K_{dp}\right)^{0.85} \qquad (2)$$

However, our results showed a clear overestimation with the radar-derived rainfall rates R(kdp), even for Group 2 rain gauges, where the highest rainfall intensities were recorded during the tropical cyclone.

Therefore, we calculated the coefficient of R(Kdp) from our radar observations. However, as noticed by the reviewer, the derived coefficients (a = 8.062, b = 0.49) fall outside the expected range for X-band radars (a = 14–20, b = 0.73–0.85) reported in Ryzhkov and Zrnić (2019). We acknowledge this discrepancy and provide the following explanations:

The coefficients of R(Kdp) relationships are highly sensitive to DSD characteristics, which vary significantly by precipitation regime (Unuma et al., 2025). Tropical cyclones exhibit distinct DSD properties compared to climatological rainfall: i) high concentrations of small and midsize raindrops; ii) relatively few large raindrops (exceed rarely 4mm); iii) elevated drop number concentrations at given reflectivity levels. These features have been documented in observational studies of Atlantic tropical cyclones (Tokay et al., 2008).

$K_{dp}$ is strongly influenced by both the size and number of drops (Timothy et al., 1999). Higher drop concentrations lead to larger $K_{dp}$ values for a given rainfall rate.

Therefore, the interpretation of R(kdp) coefficients proposed by this paper is as follow:
- Lower coefficient a (8.062 vs. 14–20): high drop concentrations producing higher Kdp during tropical cyclone.
- Lower exponent b (0.49 vs. 0.73–0.85): a weaker power-law relationship, indicating less sensitivity of R to Kdp changes, which may reflect the more uniform DSD characteristics in tropical cyclone precipitation.

Unfortunately, no disdrometer measurements were available during this event, we could not investigate the DSD.

We fully acknowledge the limited scope of our empirical derivation. Ideally, a comprehensive investigation across multiple tropical cyclone events would strengthen the robustness of TC-specific R(Kdp) relationship. However, the logistical constraints of the mobile radar deployment during this project precluded such extensive sampling (detailed below, P.8).

This finding opens promising research avenues for investigating tropical-cyclone-specific $R(K_{dp})$ relationships for X-band radar, using disdrometer data and T-matrix scattering simulations. These theoretical calculations, based on observed TC-specific DSDs, provide an independent verification of $K_{dp}$ behavior.

**3- The current implementation also raises the question of what would happen if R(Z) was derived in the same way. How would this differ from the 300, 1.5 relation applied already? The authors should justify why an empirical derivation is acceptable for one technique but not the other when it is clear both could be assessed in the same way.**

We thank the reviewer for this relevant comment. For R(Z), we rely on the Z–R relationship specifically established for tropical cyclones by Jorgenses and Wills, (1981), which is also used for operational radars in Reunion. This well-validated formulation is appropriate for our dataset. The reason why we did not calculate R(Z) from our radar observations.

For R(Kdp), given that coefficients a and b depend on radar wavelength, that X-band radar is relatively new in Réunion, and that existing R(Kdp) relationships in the literature, generally derived from larger climatological datasets, did not sufficiently match our observations, which is a specific case (tropical cyclone). For these reasons, we derived a R(Kdp) relationship directly from our dataset.

**4-** **The pre-processing of differential phase shift is interesting and thorough, yet it is not clear which aspects are novel in comparison to the cited paper by Padmanabhan (2024) which applies processing to the same radar data. The finding that the system differential phase is often azimuth and elevation dependent and requires careful assessment is useful to the wider radar community. The authors could emphasise this more.**

In this paper, we present a complete pre-processing workflow for the differential phase shift ΦDP. Our approach includes (i) noise suppression, (ii) phase unwrapping, and iii) removal of the system offset.

After these steps, we also corrected an additional issue identified by Thiruvengadam et al.(2025): radome-induced biases affecting ΦDP along specific azimuths and elevation angles. In this present paper, we applied their methodology to mitigate these later biases.

This complete pre-processing process will be clarified in the revised paper.

**5-** **To further highlight this point, they could also improve Figure 5. I'd suggest changing from line plots to either boxplots (for each azimuth) or a min-max interval with a median/mean + iq range/std deviation plotted too, this would make it clearer how much the offset varied in each sector. To make this more visible the authors could then select 3 elevations to plot which are representative of the whole volume and include individual plots as supplementary information if they believe them to be relevant.**

Thank you for the suggestion, below is a box plot representing all volumes and data used in this data study: February 1-4, representative of the entire data set. The same graph is included in the appendix, but for specific elevation (low, medium, and high elevation).

[Figure]

*Figure 3: Boxplot of the ΦDP system offset for all PPI volumes during the study period (1–4 February 2022), illustrating the azimuthal dependence of the offset. The orange line indicates the median, the black box spans the first (Q1) and third (Q3) quartiles, and the grey whiskers extend from Q1 − 1.5×IQR to Q3 + 1.5×IQR, where IQR = Q3 − Q1.*

The large boxplot indicates a high variability in the offset values, highlighting that using a single offset for all volume is not appropriate.

6- **Again, an inclusion of a representative PPI showing the impact of this azimuthal variation would be beneficial.**

The figure below shows examples of PPI at 9 ° of elevation.

[Figure]

*Figure 4: PPI at an elevation of 9° at 19 UTC on February 3, 2022: (a) raw differential phase shift, (b) corrected ΦDP (using the procedure described in this article), (c) raw reflectivity, and (d) system offset illustrating its variation along the azimuth.*

7- **The rain gauge analysis is limited in its applicability by the very small sample size, both in terms of the number of gauges and the length of time used. While it has some value in supporting the pre-existing scientific consensus it could benefit from either the inclusion of significantly more data from the X-band radars yearlong deployment on Reunion or by reframing the paper more as a case-study analysis of the cyclone itself and what the wider (in a spatial sense) dual-polarisation variables help explain about the cyclone and its interaction with the orography, which would seem to have great potential given the novel aspects of the deployment. Incorporating data from the S-band radars mentioned in the paper would also add to such a case study analysis.**

We acknowledge the limited sample used in this study.

This work is conducted within the framework of the ESPOIRS project, which involved deploying radars on multiple islands. The first year in Réunion corresponded to the radar takeover and a test-bed period. During this time, the

radar faced several technical challenges, including restrictions on emissions imposed by civil aviation, a long power outage corresponding to the summer closure of the establishment where the radar was installed, frequent internet outages, and a major failure of the air conditioning system, which was not adapted to the wet and hot tropical climate.

In addition, we experimented with various scanning strategies, regularly adjusting operational parameters such as pulse lengths, number of integrations, sampling, distance resolution, elevation angles, number of beams, rotation speeds, and sequencing between PPI and RHI scans. Consequently, the spatial and temporal resolution of the data, as well as the operating procedures, were not homogeneous throughout the deployment period.

However, during the passage of Tropical Cyclone Batsirai from 1 to 4 February 2022, the radar data are continuous, homogeneous, and collected with constant temporal and spatial resolution.

In addition, our main objective is also to correct and validate reflectivity attenuation corrections, which are essential for analysing precipitation dynamics, one of the objectives of the ESPOIRS project. By focusing on the TC Batsirai event, we can evaluate these corrections under heavy precipitation conditions.

Thanks to the operational experience gained during this period in Réunion, the scan strategy is now well established and has been consistently applied in subsequent deployments in the Seychelles (5 months) and Madagascar (2 months). However, quantitative validation at these sites is limited by sparse rain-gauge coverage, with only manual daily accumulation gauges available, preventing robust assessment of rainfall estimates.

For these reasons, this paper focuses on the Tropical Cyclone Batsirai event. This will be clearly agued in the revised paper.

Ryzhkov, A.V., Zrnic, D.S., 2019. Polarimetric Measurements of Precipitation, in: Ryzhkov, A.V., Zrnic, D.S. (Eds.), Radar Polarimetry for Weather Observations, Springer Atmospheric Sciences. Springer International Publishing, Cham, pp. 373–433. https://doi.org/10.1007/978-3-030-05093-1_10

***Specific Comments:***

**8- L33: Is there a reference for the 12h, 24h and 72h precipitation records? Can you include the actual record values too.**

Please find under this official website the reference : Quel endroit détient le record mondial de pluie ? | Météo-France :

- Cyclone Denise (1966) set a record with 1,144 mm in 12 hours and 1,825 mm in 24 hours.
- Cyclone Gamède (2007) broke the 72-hour record with 3,930 mm of rainfall.

**9-  L76. Including an indication of the total number of volume scans used would be useful.**

The total number of volume scans is 576.

**10-Figure 1: Can you add a 75km radar range ring to show the data extent. Perhaps you could also add a specific radar PPI here given the blank space to show the coverage during the cyclone passage.**

Below is the improved figure:

[Figure]

*(a) Location of the South-West Indian Ocean and track of the tropical cyclone Batsirai (TC); (b) Digital elevation model of Réunion Island showing the radar site, nearby rain gauges, the 75-km radar range, and the 20-km radius within which the radar beam is the closest (vertically) to the rain gauges. (c) Zoom over the study area. (d) PPI of raw reflectivity at 11° elevation, observed at 19 UTC on 3 February 2025.*

**11- Table 1: Can you update to include actual values used during the data collections, for example the azimuth rotation speed, the PRF and the pulse width. These are much more useful for considering the data here than the range of possibilities available for any deployment.**

Thank you for the suggestion, below is the updated table**.**

| | |
|---|---|
| **Operating frequency** | 9410 MHz ± 30 MHz |
| **Peak power** | 25 kW (12.5 kW per channel) |
| **Transmitter** | Magnetron |
| **Pulse length** | 0.75 µs |
| **Pulse repetition frequency** | 650 Hz |
| **Range resolution** | 25 m (RHI), 125 (PPI) |
| **Range** | 50 km (RHI), 75 km (PPI) |
| **Polarization** | Dual polarization (H/V) |
| **Receiver** | Dual pol. (2 independent channels) / Doppler |
| **Antenna** | 1.3 m splash plate parabolic antenna |
| **Beam width** | < 2° |
| **Antenna motion** | Volume scan (12 PPI's) |
| **Speed azimuth** | 15 °/s |
| **Speed elevation** | 10 °/s |

**12- Table 2: Almost all the data included here is also shown in Figure 2 - I'd suggest you don't need both, and Figure 2 seems to be a better visualization for the reader.**

Thank you for the suggestion. We will remove it in the revised version.

**13- L181: Is any disdrometer data available for Reunion? Could this be used here to derive specific R(Z) and R(Kdp) relations?**

Unfortunately, in the southern Reunion, no disdrometer measurements are available; only rain gauges data can be used.

**14- L189: What exactly is a maxdisplay plot? Have you manually assessed each of these to identify stratiform times or applied an algorithm. A little more detail is needed to understand your methodology.**

A maxdisplay plot shows the maximum value of the radar reflectivity along the vertical column. As the main objective of the ESPOIRS project is to study the dynamics of orographic precipitation, we produced maxdisplay plots and RHI

scans for each time step and we analyzed them. Stratiform precipitation is characterized by the presence of bright band (L185; Fabry and Zawadzki, 1995; Matrosov, 2021). The bright band is visually easy to detect; it appears as a ring of high reflectivity on the maxdisplay and PPI plots. With RHI, bright band is clearly visible at around 5 km of altitude, characterized by high reflectivity (example below). During the study period, it occurs only on the early morning of 3rd of February, when cyclone approached Reunion Island.

We didn't apply a specific algorithm to classify stratiform precipitation but used visual determination to identify the stratiform cases.

[Figure]

*Figure 5: a) Example of madisplay and b) RHI plot showing the bright band*

**15-L239: In this section the H-B implementation is described, and one of the main limitations of the method is its tendency to explode to unrealistic PIA values. In the literature many studies therefore cap the maximum PIA that can be applied. Did you do this here? If not, why not? Also how applicable are the c and d values used for this region, would you expect them to vary with the DSD?**

Yes, we set the maximum threshold to 59.0 and applied the algorithm developed in wradlib https://docs.wradlib.org/en/2.2.0/generated/wradlib.atten.correct_attenuation_hb.html#wradlib.atten.correct_attenuation_hb . For c and d, we used values from the literature for X-band radars. We didn't vary them with DSD. In addition, given the strong attenuation observed during the event, we moved directly to dual polarization techniques to correct for attenuation.

These arguments will be added in the paper discussion.

**16-L253: Here you refer to many alpha values without citing what atmospheric conditions/DSDs they were derived from. Can a more focused approach be taken?**

The table below illustrates the atmospheric conditions under which alpha values were derived:

| Koffi et al., (2014) | $\alpha$ = 0.285 dB/° | Temperature : 25°C Based on observed radar data by analyzing the slope in the scatter plots between the uncorrected horizontal reflectivity and the measured $\phi_{DP}$, method proposed by Carey et al. (2000) |
|---|---|---|
| Gamic Manufacturer | $\alpha$ = 0.28 dB/° | Atmospheric conditions are not described in the technical manual |
| Gaussiat, and Tabary (2018) | α = 0.276 dB/° | Based on observed radar data Atmospheric conditions: summer rainfall events in France Metropolitan (~46°N of latitude) |
| Meteo France, operational radars (Boumahmoud *et al.,* 2010; and cited by Figueras i Ventura *et al.,* 2012). | $\alpha$ = 0.28 dB/° | Meteo France Based on observed radar data. (The atmospheric conditions were not mentioned) |

**17- One of the limitations of linear phi is that alpha can vary along the ray, especially in the presence of hail (so called hot-spots). Is that a concern here?**

During the study period, the type of precipitation is only rain, we assume that alpha is constant along the radius.

**18- L267: What constitutes a "high gradient value"? Can you give specific thresholds that are used?**

thank you for the question. Indeed, this step is not detailed in the submitted paper. We will add the detail in revised version:

This filter removes abnormally noises from $\Phi DP$ measurements:
- ✓ First, $\Phi DP$ is normalized to $[-180°, 180°]$ to avoid phase-wrapping artifacts.
- ✓ Horizontal (range) and vertical (azimuth) gradients are computed, and pixels with gradients exceeding five times the standard deviation of the gradient field are flagged **(so called strong gradient).**
- ✓ -A spatial consistency check is then applied: pixels are removed only if they and at least two neighboring pixels (left, right, above, below) all show strong gradients. This approach eliminates noise clusters.

**19- Figure 5: See my comments in the general section. I think this is a very valuable result but the presentation can be improved to avoid the horizontal lines you see in many of the plots. Consider a hexbin or a boxplot or something else that can depict the range of values at each azimuth more clearly. Also why do you restrict this to 3 hours in particular? Does it change if using a different 3-hour windows?**

Thank you for the suggestion. We selected a 3-hour period during which continuous stratiform precipitation occurred. Stratiform precipitation is characterized by homogeneous and widespread rain, which makes the azimuthal dependence of the offset clearly visible in line plots. In contrast, during convective precipitation, the rain field is spatially heterogeneous, and the azimuthal dependence of the offset is not clearly visible in line plots. Our goal here was to show the azimuthal dependence of the system offset. In the revised version, we will replace the line plot with a boxplot, as shown in Figure 3.

**20-Figure 6: You could add an intermediate ray trace, after the removal of system phase and unfolding to show the variability before and after smoothing.**

[Figure]

*Figure 6: Raw differential phase (blue), corrected differential phase $\phi dp$ (black) and differential phase $\phi dp$ before smoothing (red), the "x" in the figure indicates the azimuth where the differential phase system offset is located (meeting the criteria described in section 2.2.2.1) : 9° scan, azimuth angle 337 on 3 February 2022 at 19:00 UTC*

**21-Figure 7 and surrounding section: This requires a lot more detail as to how you compare 6 minute Kdp with hourly gauge totals, a description of exactly what data is going onto the plot and why. For example is this from all gauges or just one. Are there thresholds applied to rain-rate or kdp?**

We thank the reviewer for this helpful comment. We agree that additional clarification was needed.

The rain-gauge data have a 6-min temporal resolution, whereas the radar $K_{dp}$ estimates are available every 10 minutes. To ensure a consistent comparison between the two datasets, both variables were aggregated to a common 30-min temporal resolution. A 30-min interval was retained only if valid measurements were available for both rainfall and $K_{dp}$.

Sincee $K_{dp}$ values close to zero may correspond either to light precipitation or to unreliable estimates, so only strictly positive $K_{dp}$ values were considered. Any interval containing missing or non-positive $K_{dp}$ values was discarded. After applying these filters and separating the test and training data, only 82 samples met the data quality criteria and were included in the analysis.

**22-L418: I would consider a NB of 0.5 to be more than a "slight tendency" to overestimate as it indicates a 50% overestimation by the radar on average. More discussion of this and the reasons for it would be beneficial.**

**L422: I'd argue that the NB results don't indicate that R(Kdp) is better at least for group 1, where all 3 gauges have a lower NB using R(Z) than R(Kdp).**

We acknowledge this important observation. For Group 1, we agree that describing a normalized bias (NB) of 0.5 as a "slight tendency" was inappropriate. Moreover, although the correlation of R(Kdp) is higher than that of R(Z), the NB remains substantial, and R(Z) therefore performs better than R(Kdp) for this group. This point has been corrected in the revised manuscript.

The overestimation of R(Kdp) is particularly pronounced for group 1, which comprises stations located near the radar, at lower altitudes. The strong orographic gradients over Réunion Island led to highly heterogeneous rainfall distributions (Figure 9), with group 1 experiencing weaker precipitation intensities than group 2. Reasons why R(kdp) perform better in group 2, regarding correlation and NB.

**23-L467: Again only correlation indicates R(Kdp) is an improvement, with NB being inconclusive. The picture is much more complicated than R(Kdp) being a significant improvement over R(Z) and this is probably the result of several factors such as the variability of attenuation, changes in VPR and changes in the accuracy of Kdp estimation at different intensities of rainfall. It's interesting here that you don't consider the use of a hybrid approach as is often implemented where R(Kdp) is used for higher rain-rates where it benefits from reduced estimations noise, less DSD variability and immunity to attenuation but where R(Z) is used at lower**

**rain-rates when attenuation is lower, Kdp is harder to estimate and the R(Kdp) relation has more DSD variability.**

We thank the reviewer for this important remark. In our initial evaluation, we indeed gave substantial weight to the correlation coefficient.

Regarding the spatial distribution of rainfall on Réunion due to orographic effects, our results show that high-elevation stations (group 2) tend to be underestimated with R(Z), primarily due to strong attenuation in regions of heavy rain. In contrast, R(Kdp) provides better estimates in these cases, as Kdp is less affected by attenuation. Conversely, at lower-elevation stations (group 1), R(Kdp) tends to overestimate rainfall.

We therefore agree with the reviewer that a hybrid estimation scheme is highly relevant for Réunion Island, where strong rainfall heterogeneity exists. Using R(Kdp) for high rain rates, where Kdp is robust and less noisy, and R(Z) for low to moderate rain rates, where Kdp becomes less reliable.

The opportunity and benefits of such an hybrid approach will be presented and argued in the revised paper.

**24-L486: "just under a year" This raises the question why the paper only focuses on one 4 day case here. Is there a complementary case study that could work alongside it, or just the potential to expand to include all the data to get a more representative set of statistics.**

We thank the reviewer for this question. As noted in Reviewer 1's comment, point 7, we provided the following explanations:

This work is conducted within the framework of the ESPOIRS project, which involved deploying radars on multiple islands. The first year in Réunion corresponded to the radar takeover and a test-bed period. During this time, the radar faced several technical challenges, including restrictions on emissions imposed by civil aviation, a long power outage corresponding to the summer closure of the establishment where the radar was installed, frequent internet outages, and a major failure of the air conditioning system, which was not adapted to the wet and hot tropical climate.

In addition, we experimented with various scanning strategies, regularly adjusting operational parameters such as pulse lengths, number of integrations, sampling, distance resolution, elevation angles, number of beams, rotation speeds, and sequencing between PPI and RHI scans. Consequently, the spatial and temporal resolution of the data, as well as the operating procedures, were not homogeneous throughout the deployment period.

However, during the passage of Tropical Cyclone Batsirai from 1 to 4 February 2022, the radar data are continuous, homogeneous, and collected with constant temporal and spatial resolution.

In addition, our main objective is also to correct and validate reflectivity attenuation corrections, which are essential for analysing precipitation dynamics, one of the objectives of the ESPOIRS project. By focusing on the TC Batsirai event, we can evaluate these corrections under heavy precipitation conditions.

Thanks to the operational experience gained during this period in Réunion, the scan strategy is now well established and has been consistently applied in subsequent deployments in the Seychelles (5 months) and Madagascar (2 months). However, quantitative validation at these sites is limited by sparse rain-gauge coverage, with only manual daily accumulation gauges available, preventing robust assessment of rainfall estimates.

For these reasons, this paper focuses on the Tropical Cyclone Batsirai event.

**Appendix**

[Figure]

Boxplots of differential phase system offset at 4°, 15° and 19°elevation, showing its variation along the azimuth.

**References**

Beard, K. V. and Chuang, C.: A New Model for the Equilibrium Shape of Raindrops, J. Atmospheric Sci., 44, 1509–1524, https://doi.org/10.1175/1520-0469(1987)044%253C1509:ANMFTE%253E2.0.CO;2, 1987.

Figueras I Ventura, J., Boumahmoud, A., Fradon, B., Dupuy, P., and Tabary, P.: Long-term monitoring of French polarimetric radar data quality and evaluation of several polarimetric quantitative precipitation estimators in ideal conditions for operational implementation at C-band, Q. J. R. Meteorol. Soc., 138, 2212–2228, https://doi.org/10.1002/qj.1934, 2012.

Hitschfeld, W. and Bordan, J.: ERRORS INHERENT IN THE RADAR MEASUREMENT OF RAINFALL AT ATTENUATING WAVELENGTHS, J. Atmospheric Sci., 11, 58–67, https://doi.org/10.1175/1520-0469(1954)011%253C0058:EIITRM%253E2.0.CO;2, 1954.

Jorgenses, D. P. and Wills, P. T.: A Z-R relationship for Hurricanes, J. Appl. Meteorol., 1981.

Thiruvengadam, P., Lesage, G., Ramanamahefa, A. V., and Van Baelen, J.: Mitigating radome-induced bias in X-band weather radar polarimetric moments using an adaptive discrete Fourier transform algorithm, Atmospheric Meas. Tech., 18, 1185–1191, https://doi.org/10.5194/amt-18-1185-2025, 2025.

Timothy, K. I., Iguchi, T., Ohsaki, Y., Horie, H., Hanado, H., and Kumagai, H.: Test of the Specific Differential Propagation Phase Shift (KDP) Technique for Rain-Rate Estimation with a Ku-Band Rain Radar, J. Atmospheric Ocean. Technol., 16, 1077–1091, https://doi.org/10.1175/1520-0426(1999)016%253C1077:TOTSDP%253E2.0.CO;2, 1999.

Tokay, A., Bashor, P. G., Habib, E., and Kasparis, T.: Raindrop Size Distribution Measurements in Tropical Cyclones, Mon. Weather Rev., 136, 1669–1685, https://doi.org/10.1175/2007MWR2122.1, 2008.

Tridon, F.: Mesure des précipitations à l'aide d'un radar en bande X non-cohérent à haute résolution et d'un radar en bande K à visée verticale. Application à l'étude de la variabilité des précipitations lors de la campagne COPS, 199, 2011.

Unuma, T., Yamauchi, H., Kato, T., Umehara, A., Hashimoto, A., Adachi, A., and Nagumo, N.: Characteristics of Raindrop Size Distribution Using 10-year Disdrometer Data in Eastern Japan, J. Meteorol. Soc. Jpn. Ser II, 103, 219–232, https://doi.org/10.2151/jmsj.2025-011, 2025.